# In Vitro and In Vivo Characterization of PLLA-316L Stainless Steel Electromechanical Devices for Bone Tissue Engineering—A Preliminary Study

**DOI:** 10.3390/ijms22147655

**Published:** 2021-07-17

**Authors:** Mariana V. Branquinho, Sheila O. Ferreira, Rui D. Alvites, Adriana F. Magueta, Maxim Ivanov, Ana Catarina Sousa, Irina Amorim, Fátima Faria, M. H. V. Fernandes, Paula M. Vilarinho, Ana Colette Maurício

**Affiliations:** 1Veterinary Clinics Department, Abel Salazar Biomedical Sciences Institute (ICBAS), University of Porto (UP), Rua de Jorge Viterbo Ferreira, n° 228, 4050-313 Porto, Portugal; m.esteves.vieira@gmail.com (M.V.B.); ruialvites@hotmail.com (R.D.A.); anacatarinasoaressousa@hotmail.com (A.C.S.); 2Animal Science Studies Centre (CECA), Agroenvironment, Technologies and Sciences Institute (ICETA), University of Porto (UP), Rua D. Manuel II, Apartado 55142, 4051-401 Porto, Portugal; 3Department of Materials and Ceramic Engineering, CICECO, Aveiro Materials Institute, University of Aveiro, 3810-381 Aveiro, Portugal; sheila_oliveira_93@hotmail.com (S.O.F.); adrianamagueta@ua.pt (A.F.M.); maxim.ivanov@uc.pt (M.I.); helena.fernandes@ua.pt (M.H.V.F.); paula.vilarinho@ua.pt (P.M.V.); 4Department of Pathology and Molecular Immunology, Abel Salazar Biomedical Sciences Institute (ICBAS), University of Porto (UP), Rua Jorge Viterbo Ferreira, n° 228, 4050-313 Porto, Portugal; irinamorim@hotmail.com (I.A.); fatimafaria10@yahoo.com.br (F.F.); 5Institute of Research and Innovation in Health (i3S), University of Porto (UP), Rua Alfredo Allen, 4200-135 Porto, Portugal

**Keywords:** PLLA, SS316L, bone regeneration, biomaterials, cytocompatibility, biocompatibility

## Abstract

Bone injuries represent a major social and financial impairment, commonly requiring surgical intervention due to a limited healing capacity of the tissue, particularly regarding critical-sized defects and non-union fractures. Regenerative medicine with the application of bone implants has been developing in the past decades towards the manufacturing of appropriate devices. This work intended to evaluate medical 316L stainless steel (SS)-based devices covered by a polymer poly (L-lactic acid) (PLLA) coating for bone lesion mechanical and functional support. SS316L devices were subjected to a previously described silanization process, following a three-layer PLLA film coating. Devices were further characterized and evaluated towards their cytocompatibility and osteogenic potential using human dental pulp stem cells, and biocompatibility via subcutaneous implantation in a rat animal model. Results demonstrated PLLA-SS316L devices to present superior in vitro and in vivo outcomes and suggested the PLLA coating to provide osteo-inductive properties to the device. Overall, this work represents a preliminary study on PLLA-SS316L devices’ potential towards bone tissue regenerative techniques, showing promising outcomes for bone lesion support.

## 1. Introduction

Orthopedic injuries represent an important clinical situation, impairing the well-being of individuals worldwide, also associated with great financial restrains [1]. The bone tissue has a limited self-renewal capacity, furthermore, critical sized bone defects (from up to 2 cm, varying on the anatomical site) or non-unions, are unable to selfheal, thus requiring further medical intervention [2,3,4]. Nonetheless, non-critical sized defects are often associated with a long-term recovery period, not always associated with biologically functional outcomes. Several situations can culminate in tissue loss, from fractures, tissue removal, such as tumors, and other diseases, as congenital situations. Current treatments include bone grafting techniques, with resort to allograft and autografts, for massive tissue loss situations, as well as bone fixation with inert metallic biomaterials [2]. Tissue engineering for bone regeneration is often related to 3D printed scaffolds (polymers, ceramics, and composite materials), mimicking the bone’s structure and extracellular matrix (ECM), thus facilitating new bone formation. On the other hand, when considering bone defects’ support and stabilization, tissue engineering techniques rely on structural biomaterials, capable of supporting physical load motion and resistance to the fracture, being the most common based on metals and their alloys [5,6]. Among them, the most frequently used are stainless steel (SS), cobalt-based, and titanium-based biomaterials. The SS class are iron (Fe)-based alloys with varying percentages of chromium (Cr) and nickel (Ni) and are associated with a great ability to bear significant loads. In this work, a medical SS device with the identification code 316L was employed (Fe (69%), Cr (18%); Ni (10%) and molybdenum (Mo) (3%)). The Cr promotes the passivation ability, while the Mo enhances the corrosion resistance of the devices. Considering the biomaterial’s classification, a SS316L device is characterized as an austenitic type, being widely applied in the medical field, mainly for short-term implants, as they present insufficient corrosive resistance for long-term applications [2,5]. Nonetheless, these biomaterials are extensively applied for short-term internal fixation (plates, screws, i.a.), due to their prompt availability, low cost, good fabrication properties, and toughness resistance. When designing devices for short-term fixation, some criteria must be considered, as the devices’ biocompatibility, noncorrosive nature, advantageous mechanical properties, load resistance, and in some cases, osteointegration potential [2]. As for the devices’ biocompatibility, it has been described as relatively good for SS316L devices, probably due to the presence of Cr and Ni, both associated to some toxicity levels, and in the long-term characterized as a fatigue corrosion phenomenon (metallosis). Care must be taken when considering these alloys, as their weight bearing capacity outperforms the intrinsic bone loading potential, thus leading to an exclusively load bearing on the device. Consequently, the surrounding bone, without weight-bearing stimulation will reduce bone density (osteopenia) [2], leading to a phenomenon described as stress shielding. These devices are associated with poor osteointegration properties, a critical characteristic when considering the devices’ stability in situ [2,6]. With this regard, this work intended to modify a SS316L device, by coating the surface with a biocompatible layer, as to promote a mechanically stable connection between the bone and the device, and thus, favoring osteointegration. Various biocompatible coating substances have been widely applied for bone regeneration, as hydroxyapatite, graphene-oxide, calcium phosphates (CaP), zirconium titanate (ZrTiO4), and poly (L-lactic acid) (PLLA) [7,8]. Among them, the PLLA attracts significant attention being a synthetic semicrystalline piezoelectric polymer associated with controlled degradation rates, adjustable physical properties, and with reported promising biocompatibility outcomes [2,6,9,10,11]. Several studies have observed an acceleration in bone regeneration on PLLA devices. Bone tissue is known to possess an intrinsic electromechanical activity, i.e., a piezoelectric nature [9], due to the presence of collagen and hydroxyapatite crystals [12,13,14]. Furthermore, these devices have the ability to respond to external stimulus (e.g., temperature, stress, electrical field), creating a piezoelectric effect [15,16,17,18], that increases their possible potential towards promoting bone tissue regeneration. Previous works have entailed PLLA-based devices characterization and cell interaction with promising results towards bone regeneration [9,10,19]. In the scope of this work, devices of SS316L were coated with PLLA using a highly efficient silanization procedure, previously described by Magueta et al. [20]. PLLA coating films were submitted to a thermal treatment, as to increase the crystallization degree, as it has been shown to enhance cellular adhesion [18,21]. Devices were further characterized in vitro, assessing on their cytocompatibility properties, and in vivo, by subcutaneous implantation on a rat animal model, following ISO 10993-6:2016 guidelines, assessing on the devices’ biocompatibility. This work presents preliminary results, on the PLLA coating of SS316L devices’ performance towards bone tissue regeneration techniques.

## 2. Results

### 2.1. Characterization of the Devices

The morphology of the surface of SS/SIL/PLLA was studied by scanning electronic microscopy (SEM) and the micrographs are presented in Figure 1a, clearly showing the formation of spherulites with a well-defined grain boundary. The mean diameter of spherulite and the corresponding standard error determined was 88.9 ± 1.7 μm. Big spherulites with a diameter size distribution ranging from 19.4 to 270.4 μm were observed. The crystallographic structures and compounds of SS/SIL/PLLA are identified in the diffractogram presented in Figure 1b. Based on the JCPDS-PDF base [C3H5O3)n, 00-054-1917], peaks at 16.6 and 19.0° were ascribed to PLLA α form, being (110)/(200) and (203)/(113) the corresponding crystallographic plans, respectively. The thermal properties of PLLA films were studied by differential scanning calorimetry (DSC) (Figure 1c). During heating, the glass transition (T_g_) and the melting temperature (T_m_) were 60.4 and 179.4 °C, respectively. During cooling, the crystallization temperature (T_c_) was 73.0 °C.

Using the Origin Pro 8 software, the peak area of T_m_ was calculated in the plot of heat flow (J/s·g) as a function of time (s) [22]. The specific enthalpy of fusion (J/g) of the PLLA film (${\Delta H}_{f}^{}$) was −69.80 J/g. Considering the enthalpy of fusion of 100%, crystalline PLLA samples with α-crystals (${\Delta H}_{f}^{0}$) with the value of −143.00 J/g [23], presented a degree of crystallization of 48.8%. Images of both devices are presented in Figure 1d,e, corresponding to SS/SIL and SS/SIL/PLLA, respectively.

### 2.2. Cytocompatibility Assessment

The Presto Blue^TM^ viability assay was performed in SS/SIL (gold standard) and SS/SIL/PLLA samples. A group with seeded cells but with no biomaterial was considered as a control of the cell population health and normal behavior in culture, growth, and proliferation. Corrected absorbance values were obtained for each time-point (24, 72, 120, and 168 h) and are presented in Figure 2 (left panel) and Table 1. The % of viability inhibition, normalized to the SS/SIL group, is presented on the right panel of Figure 2 and Table 2.

The cytocompatibility assessment for the devices was performed by the Presto Blue™ viability assay and with the use of hDPSCs. Although the reagent used is not referred in the ISO 10993-5:2009 guidelines, the PrestoBlue™ was employed for this assay, as it allows to perform live-cell assays, in contrast with, for example, the MTT assay, that only allows for endpoint assays. Thus, using this method, the exact same cell population can be evaluated over time in addition to using fewer devices. As such, this assay is less time consuming and implies an inferior financial investment. Nevertheless, the PrestoBlue™ has a comparable performance with other cell viability reagents [24]. Despite not being listed on the guidelines, this assay was adapted from ISO 10993-5:2009 “Biological evaluation of medical devices”—Part 5—“Test for in vitro cytotoxicity”. The data analyses were performed according to the manufacturing instructions. Viability inhibition, when compared to the SS/SIL (gold standard) group, superior to 30% was considered a cytotoxic effect, according to annex C of the referred guideline. The cell line hDPSCs was selected due to these cells’ pre-established regenerative potential towards the osteogenic lineage [25,26,27], complying with the purpose of testing the biological potency of these devices towards bone tissue regeneration. The results show a normal cell proliferation and growth rate, when observing the control group, seeded directly on the well bottom, with no device, thus, confirming this assay viability for further statistical interpretation. The SS/SIL/PLLA devices presented overall a superior cytocompatibility performance, when compared with the gold standard SS/SIL group. The enhanced cytocompatibility outcomes can thus be associated with the PLLA coating, as it has been proposed by other groups working with PLLA devices [9,19,28,29]. As shown in the % viability inhibition graph in Figure 2 (right panel), the SS/SIL/PLLA group can be classified as non-cytotoxic, as the % of viability inhibition did not exceed the 30% pre-established limit throughout the assay, according to the upper mentioned guideline. In fact, this group outperformed the SS/SIL group, in terms of in vitro cytocompatibility, with an exception for the early 24 h timepoint, which can be caused by a delayed attachment from the cells to the PLLA coated surface.

### 2.3. Osteogenic Differentiation Assay

The devices’ potential to promote osteogenic differentiation was assessed for SS/SIL and SS/SIL/PLLA by the ARS solution protocol after 21 days, as described in previous works [30], through the detection of extracellular calcium deposition and intracellular presence of mineralized particles. ARS was extracted from the cellular monolayer and detected at 405 nm absorbance value. The results are presented in Figure 3 and Table 3 and Table 4.

The hDPSCs were used for this assay, as for the cytocompatibility assessment. The results shown in Figure 3 demonstrate the PLLA coated devices to enhance mineral deposition in the group supplemented with osteodifferentiation media, when compared to the SS/SIL devices, hence suggesting PLLA coating to promote osteodifferentiation. Sustaining this hypothesis, in the undifferentiated group, the SS/SIL/PLLA devices presented superior outcomes, suggesting the PLLA coating to induce spontaneous osteodifferentiation, and as such, to possess osteo-inductive properties. A control group with no devices was included in this assay, similarly to the previous assay, as to assess on the cell population health and normal behavior in standard and in differentiation supplementation conditions. Direct comparisons between the devices’ groups and this control group should be taken carefully, as, for example, the seeding areas are different (1 cm^2^ for the devices and 1.9 cm^2^ for the control group), thus explaining to some extent, the differences observed between these results.

### 2.4. Scanning Electronic Microscopy (SEM)

Following in vitro viability assessment and osteogenic differentiation, SS/SIL and SS/SIL/PLLA devices were further processed for SEM and energy dispersive X-ray spectroscopy (EDS) analysis. Devices without (unseeded) and with undifferentiated and differentiated cells were observed by SEM on different magnifications. The results are presented in Figure 4, Figure 5 and Figure 6.

SEM imaging allowed the confirmation of cell presence (fibroblast-like), with adequate adhesion, normal morphology, and layer formation for both devices’ groups. Small differences were observed regarding cell appearance, probably due to the topographic differences between both devices. Differentiated cells presented some extent of spontaneous cell aggregation, a behavior previously described for MSCs cultured in low serum supplemented medium, and thus considered normal [31]. This phenomenon can also be linked to an increase potential of these cells to differentiate into the osteogenic lineage, as aggregates have been widely reported to present superior differentiation potential compared to the 2D cell culture [32,33]. The EDS analysis, presenting the spectrum of detected energies, allowed differentiating between both devices’ surface element composition. In the non-coated group (SS/SIL), Fe, Cr, Ni, and Mo are detected, as expected due to the devices’ composition. Considering the PLLA coated group (SS/SIL/PLLA), the spectrum detects carbon (C) and oxygen (O), and the SS/SIL elements are no longer detected. This analysis allowed confirmation of an adequate PLLA coating.

### 2.5. In Vivo Biocompatibility Assessment of Implantable Devices in Subcutaneous Tissue

A thorough biocompatibility assessment for SS/SIL/PLLA devices was addressed, according to ISO 10993-6:2016 guidelines for Biological evaluation of medical devices, Part 6: Tests for local effects after implantation. Devices were subcutaneously implanted at the dorsum of the animals, as descried in annex B: “Test methods for implantation in subcutaneous tissue” of the referred guideline. Implanted samples were analyzed at 3, 7, 15, and 30 days post-implantation time, and a semi-quantitative scoring system was applied, according to annex E: “Examples of evaluation of local biological effect after implantation”. SS/SIL devices were considered for each animal and were used as negative control for each timepoint. The scoring system implied a global histological scoring of each sample, with a detailed histopathologic evaluation of the biological tissue response, including fibrosis extent, tissue morphology changes, inflammatory cell types, necrosis presence and extent, neovascularization, fatty infiltration, and other relevant parameters, according to the guideline. The results are presented in Table 5 and Figure 7.

Macroscopical evaluation of the subcutaneous tissue revealed no infection, inflammation, nor hemorrhage. The scoring system was obtained by subtracting the global histological score of the SS/SIL devices for each timepoint. Overall, microscopical samples presented minimal fibrosis and a broad brand of capillaries with supporting structures, at every timepoint (Figure 8). The presence of polymorphonucleated (PMN) cells was detected at each timepoint for both samples but decreased along with the recovery period. Necrosis, as well as giant cells were only seldomly identified. Mononuclear inflammatory cells were greatly detected, when compared to PMN cells, for all groups and recovery periods, except for the 3 days timepoint. According to the ISO 10993-6 scoring system, the SS/SIL/PLLA device was classified as “slight reaction” at 3 days post-implantation time, and as “minimal to no reaction” at 7, 15, and 30 days post-implantation time, compared to the gold standard SS/SIL devices. No relevant alterations were noted in the microscopical evaluation of the different organs. Considering the previous mentioned guideline, SS/SIL/PLLA devices can be classified as biocompatible.

## 3. Discussion

This work intended to evaluate the in vitro and in vivo performance of polymer functional metal devices, envisioning large bone defects stabilization. No ideal implant has yet been established, each material presents its own advantages and disadvantages. Considering the regenerative purpose in view, the combination of different materials, may overcome the prementioned impairments [34]. In this regard, the authors propose, with this work, the hypothesis that the combination of the mechanical properties of the SS, along with the biocompatible characteristics of the PLLA coating, could improve the devices’ potential towards the stabilization of large bone defects (critical sized bone defects or non-union fractures). SS/SIL/PLLA devices with well-defined spherulites were produced and characterized. Previous works from our group have shown that the electromechanical performance and stability of the polarization are favored by a more ordered crystalline form of PLLA [9,35,36]. However, too high crystallinity degrees have been associated with inhibition of osteoblast cells proliferations [37]. Thus, in this work, a moderately high crystallinity degree was considered (48.8%) [20]. Devices were further assessed as to their cytocompatibility and osteogenic differentiation ability with hDPSCs cell population, compared to gold standard SS/SIL devices. Mesenchymal stem cells (MSCs) have been widely applied in regenerative medicine [38], including bone tissue engineering, due to their proliferative and differentiation capabilities [26,39,40]. The hDPSCs potential towards bone tissue regeneration has been previously described [25,26,27]. These cells have recognized the potential to promote mineral deposition and further osteodifferentiation enhancement, both in vitro and in vivo and have been successfully applied in bone tissue engineering, associated with 3D scaffolds [25,26,27,41,42]. The validity of these cells’ application for in vitro and in vivo studies towards bone tissue regeneration has been previously established and accepted [43,44]. Furthermore, envisioning clinical application, they represent a non-invasive and easily accessible cell source [26]. Moreover, these cells can be easily cryopreserved, present high proliferation rates in culture, and are capable of secreting bioactive factors, thus enhancing their regenerative ability [26,41,42]. Regarding the cytocompatibility assay, SS/SIL/PLLA devices performed non-cytotoxic effects on cell viability and proliferation, following the ISO 10993-5:2009 criteria. According to the results obtained, the SS/SIL/PLLA devices presented superior viability outcomes, thus suggesting the PLLA coating to enhance cell proliferation and viability, when compared with the uncoated SS/SIL devices (Figure 2, Table 1 and Table 2). Due to the non-transparent nature of the devices, a morphologic qualitative assessment was performed by SEM, with both devices (SS/SIL and SS/SIL/PLLA) presenting cellular monolayer formation, with normal cell morphology and attachment (Figure 4 and Figure 5). The SS316L potential as a non-permanent device for bone remodeling has been widely accepted, and previous groups have been applying this alloy alone or in combination with other biomaterials, showing positive results [45,46]. Previous groups have reported promising outcomes when applying PLLA polymers in bone scaffolding, showing enhanced osteogenesis and proliferation of cell populations from various sources [47,48], as well as positive in vivo outcomes [49]. Further studies have reported this biomaterial in combination with metal substrates to enhance bone remodeling [49]. Osteogenic differentiation extension was assessed by Alizarin Red S, with the SS/SIL/PLLA devices presenting greater mineral deposition, when compared to the SS/SIL group (Figure 3, Table 3 and Table 4). Moreover, SS/SIL/PLLA devices showed superior mineral deposition in the undifferentiated group, thus, indicating that the PLLA coating can present, to some extent, osteo-inductive properties [2], by promoting spontaneous differentiation towards the osteogenic lineage. Thus, this result can indicate that the PLLA coating on SS316L devices provide the implant with an intrinsic capacity to stimulate osteogenesis. In line with these outcomes, a previous study applying PLLA devices with different components reported them to present osteoconductive and osteo-inductive properties in vitro [47]. Others have successfully assessed on PLLA based composites to present positive osteogenic differentiation outcomes [49]. Nonetheless, their works intended to evaluate PLLA on 3D scaffolds for bone substitution, and not as a mechanically functional device for bone lesion support. Further studies compared cytocompatibility and osteogenic differences between SS316L devices and SS316L alloys with an altered surface, presenting positive osteo-inductive outcomes for the latter, similarly to the results obtained in this work with the PLLA coating [50,51,52]. Regarding the biocompatibility assessment in the rat model, the absence of host tissue inflammatory response, and following the referred guidelines, SS/SIL/PLLA devices can be considered biocompatible and suitable for a short-term in vivo application. In addition to being biocompatible, SS/SIL/PLLA outperformed the SS/SIL devices, except in the 3 days post-implantation period (slight reaction). Thus, suggesting the PLLA coating to overall enhance the biocompatibility of the devices. Following the in vivo biocompatibility assessment, an in vivo bone lesion scale-up animal model (sheep) is envisioned, considering these devices and additional ones with further physical alterations on their surfaces. The non-critical and critical bone defect will be considered, on the femur and on the iliac crest, respectively, according to previous bone lesion models [3,25]. Devices will be prepared as screws and plates. The authors believe that these devices, along with further improvements to their surface, present strong regenerative characteristics towards bone lesion stabilization and support.

## 4. Materials and Methods

### 4.1. Devices’ Preparation

A metallic substrate with 10 × 10 × 0.38 mm^3^, AISI 316 Fe/Cr18/Ni10/Mo3 (FF210340 Goodfellow Cambridge Limited, Cambridge, UK) was employed as a matrix material for the devices manufacturing. Substrates were heat-treated at 500 °C for 2 h in air and chemically functionalized in a 1% (*v*/*v*) toluene solution of (3-Aminopropyl) trimethoxysilane (APTES) for 60 min in air, by a silanization process, as described by Magueta et al. [20]. Hereafter, this group is named as SS/SIL samples, and is considered the gold standard device [53]. Furthermore, a PLLA solution of 2.5 wt% was prepared by dissolving PLLA pellets (Purasorb^®^ PL 38, Purac biochem, Gorkum, Netherlands) at 80 °C for 2 h in 1.4 dioxane (99.8%, Sigma-Aldrich, St. Louis, Missouri, EUA) in air atmosphere. Approximately, 100 μL of this solution was deposited by spin coating (Chemat Technology, Inc., Northridge, CA, USA, Spin Coater KW-4A model) for 30 s at 3500 rpm on a group of pre-silanized substrates. The process was repeated 3 times to achieve a three-layer film. To promote the PLLA crystallization, the silanized substrates covered with PLLA films were submitted to a heat treatment at 180 °C for 3 min, followed by 45 min at 120 °C. The temperatures were based on the glass transition temperature (T_g_) and melting temperature (T_m_) of PLLA, ~60 and ~180 °C, respectively. These devices were named as SS/SIL/PLLA.

### 4.2. Characterization of the Devices

The morphological, structural, and thermal analyses of the SS/SIL/PLLA samples was performed, although a detailed characterization of both devices has already been conducted by Magueta et al. [20]. The morphology of PLLA films was studied by Scanning Electron Microscopy (SEM) (Hitachi SU-70, Hitachi High Tech, Schaumburg, IL, USA) under the electron acceleration field of 5 kV. The average spherulite diameter was calculated by measuring 328 spherulites using the ImageJ software, National Institutes of Health, Bethesda, Maryland, USA. Crystallographic planes were studied by X-ray diffraction (XRD). The experiments were performed at room temperature with a scan range of 5°< 2θ < 100°, using the Panalytical Xpert PRO3 equipment, Malvern, Almelo, Netherlands. To determine the degree of crystallization of PLLA films, a Differential Scanning Calorimetry (DSC) (Shimadzu, Kyoto, Japan, DSC-60) calibrated with a standard indium and in air atmosphere was employed. The experiments were repeated two times at a heating/cooling rate of 10 °C/min from room temperature to 200 °C under airflow conditions. The degree of crystallinity ($X_{C}$) was estimated from Equation (1), where ${\Delta H}_{f}$ is the specific enthalpy of fusion (J/g) of the sample determined from the peak area of T_m_ in the DSC graphic, and ${\Delta H}_{f}^{0}$ is the enthalpy of fusion of 100% crystalline material [54].
(1)$$X_{C}\;\left(\text{\%}\right)=\frac{{\Delta H}_{\text{f}}}{{\Delta H}_{\text{f}}^{0}}\times 100$$

### 4.3. In Vitro Assays

#### 4.3.1. Cell Culture and Maintenance

Human Dental Pulp stem/stromal cells (hDPSCs) obtained from AllCells, LLC, Alameda, CA, USA (Cat. DP0037F, Lot no. DPSC090411-01) were maintained in MEM α, GlutaMAX™ Supplement, no nucleosides (Gibco, 32561029), supplemented with 10% (*v*/*v*) fetal bovine serum (FBS) (Gibco, A3160802), 100 IU/mL penicillin, 0.1 mg/mL streptomycin (Gibco, 15140122), 2.05 µm/mL amphotericin B (Gibco, 15290026), and 10 mM HEPES buffer solution (Gibco, 15630122), Thermo Fisher Scientific, Waltham, MA USA. All the cells were maintained at 37 °C, 80% humidified atmosphere, and 5% CO_2_ environment. A characterization study of these cells is described by Campos et al. [25].

#### 4.3.2. Cytocompatibility Assessment

The cytocompatibility between the cellular system and the devices was assessed by the Presto Blue™ assay, as described by Alvites et al. [55]. The Presto Blue™ is a commercially available, ready-to-use, water-soluble preparation, and allows a live-cell assay. Cell viability assessment is based on cellular permeability to the resazurin-based solution. The latter functions as a cell viability indicator, as viable cells reduce the phenoxazine dye (resazurin), resulting in color modification that is quantitively measured over time by ultraviolet-visible spectrophotometry. Prior to the in vitro and in vivo assessments, devices were sterilized by ultraviolet light for 20 min at both sides. The hDPSCs at passage 4 were seeded over the devices at a density of 7000 cells per cm^2^ in a non-adherent 24 well plate, in a low volume suspension, allowing the cells to adhere to the devices rather than the well bottom. After 1 h of incubation period, seeded wells were concealed with a complete medium and incubated overnight at 37 °C, 80% humidified atmosphere, and 5% CO_2_ environment. An adherent 24-well plate was considered for the control group, without devices placed on the well bottom. For the Presto Blue™ assessment, at every time-point (24, 72, 120, and 168 h), the culture medium was removed from each well and replaced by a fresh complete medium, with 10% (*v*/*v*) of 10× Presto Blue™ cell viability reagent (Invitrogen, A13262, Thermo Fisher Scientific, Waltham, MA USA). Cells were incubated for 60 min at 37 °C, 5% CO_2_, and 80% humidified atmosphere. The supernatant medium was collected and transferred to a 96-well plate and absorbance was read at 570 and 595 nm. Wells were further washed with Dulbecco’s phosphate-buffered saline solution (DPBS, Gibco, 14190169) until Presto Blue™ residues were removed and the fresh culture medium was finally added to each well. For this assay, both devices and a control group were considered, and for each group, blank wells were included, without cell seeding, as shown in Figure 9. The Presto Blue™ wavelength for excitation is 570 and 595 nm for emission. For each well, the value obtained at 595 nm was subtracted from the value obtained for 570 nm (normalized value). The corrected absorbance for each experimental well (only considering seeded wells) was further obtained by the subtraction of the average of the blank wells from the normalized values of the respective sample group. Absorbance values were measured in triplicates with a Multiskan™ FC Microplate Photometer (Thermo Scientific™, 51119000, Thermo Fisher Scientific, Waltham, MA USA). Data were further processed and normalized to the mean of the gold standard group, and presented in % of viability inhibition, compared to the gold standard group.

#### 4.3.3. Osteogenic Differentiation Assay

The osteogenic differentiation assay was performed for the same three groups, similarly to the cytocompatibility assay, as described in previous works [55,56]. Briefly, cells were seeded over the devices as described for the cytocompatibility assessment. After achieving 80% confluency (approximately 3 days of incubation period), cells were transitioned to specific formulated Osteogenesis media (StemPro™ Osteogenesis Differentiation Kit, Gibco A1007201, Thermo Fisher Scientific, Waltham, MA USA). Control (undifferentiated) wells were maintained on standard culture media absent on osteogenic supplements, for each group. The media was changed every 3 days. The Alizarin Red S (ARS) assay (TMS-008-C, Merck KGaA, Darmstadt, Germany) was employed to assess for osteogenic differentiation after 21 days, for semi-quantitative analysis of the osteogenic differentiation process, as previously described in previous works [30,56,57]. Briefly, cells were fixated in 4% formaldehyde (Merck KGaA, Darmstadt, Germany, 100496), stained with a 40 mM ARS solution and incubated under gentle agitation for 30 min. Following supernatant removal, wells were washed with diH_2_O complete dye removal from the supernatant. A qualitative assessment was not achievable, due to the non-transparent nature of the devices. A semi-quantitative analysis was performed, by adding a 10% acetic acid solution (ARK2183, Sigma-Aldrich, St. Louis, MO, USA) to the wells, which were further scraped, allowing cells and mineral deposits collection. Individual collection samples were placed on an 85 °C water bath for 10 min, and were further placed on ice for 5 min. Following centrifugation, absorbance values are 405 nm and were taken in triplicates in a Thermo Scientific™ Multiskan™ FC Microplate Photometer.

#### 4.3.4. Scanning Electronic Microscopy (SEM)

Following cytocompatibility studies, seeded and unseeded devices from each group were collected and fixated for SEM, based on the UtahState University Biological Sample Fixation for SEM. Briefly, devices were washed 3 times with 0.1 M HEPES (Merck KGaA, Darmstadt, Germany, PHG0001) buffer and fixated with 2% glutaraldehyde (Merck KGaA, Darmstadt, Germany, G5882) buffered solution overnight. After that, the samples were rinsed with 0.1 M HEPES buffer 3 times, for 5 min each and under gentle agitation. Thereafter, a crescent series of alcohol were employed for dehydration (50%, 70%, 95%, and 99%) each 2–3 times and over 10–15 min. Finally, samples were impregnated in a crescent series of hexamethyldisilazane (HMDS)-alcohol solution (1:2; 1:1; 2:1) until soaked in a 98% HMDS (Alfa Aesar, A15139, Thermo Fisher Scientific, Waltham, MA, USA) solution for 15 min, 3 times. HMDS was then removed and the remaining residues were left to evaporate overnight on an air flow chamber. The SEM analysis and energy dispersive X-ray spectroscopy (EDS) exam were performed using a high resolution (Schottky) Environmental Scanning Electron Microscope with X-Ray Microanalysis and Electron Backscattered Diffraction analysis: Quanta 400 FEG ESEM/EDAX Genesis X4M, Thermo Fisher Scientific, Waltham, MA USA, operating in a high vacuum mode at an accelerating voltage of 15 kV SEM. Devices were coated with gold/palladium for 80 s and with a 15 mA current. Regarding the samples nature and whenever necessary, electron backscatter diffraction (EBSD) images were obtained.

### 4.4. In Vivo Biocompatibility Assessment of Implantable Devices in Subcutaneous Tissue

Animal testing procedures were in conformity with the Directive 2010/63/EU of the European Parliament and the Portuguese DL 113/2013. All the procedures were approved by the ICBAS-UP Animal Welfare Organism of the Ethics Committee (ORBEA) and by the Veterinary Authorities of Portugal (DGAV). Humane end points were followed in agreement with the OECD Guidelines (2000). The in vivo biocompatibility assessment was performed in adult male Sasco Sprague-Dawley rats (Charles River, Barcelona, Spain) weighing 250–300 g, as described in previous works [55,58]. Animals were housed with controlled temperature and humidity and 12–12 h light/dark cycles. Animals were fed with standard chow and water ad libitum. The surgical procedure implies intraperitoneally administration of anesthesia Xylazine/Ketamin (Rompun^®^, Bayer AG, Leverkusen, Germany/Imalgène 1000^®^, Merial, Lyon, France; 1.25/9 mg per 100 g b.w.), and aseptic skin preparation. Up to four 15–20 mm long linear incisions were made paired along the dorsum. Devices were implanted subcutaneously, skin and subcutaneous tissues were sutured, and animals recovered and returned to their housing groups. Sham groups were also considered, with surgical access to the site, but no device implantation. At 3, 7, 15, and 30 days after surgery, animals were subjected to deep anesthesia, and consequently euthanized, by lethal intra-cardiac injection (Eutasil^®^ 200 mg/mL, CEVA Sante Animale, Libourne, France 200 mg/kg b.w.). Collection of skin and subcutaneous tissues from the implant area was performed and samples were fixed in 4% formaldehyde (Merck KGaA, Darmstadt, Germany, 100496). Fixed samples were routinely processed for histopathological analysis, with Haematoxylin-Eosin (H&E) staining of 3 μm-thin sequential sections. Evaluation of the sections was performed with a Nikon microscope (Nikon, Amstelveen, The Netherlands, Eclipse E600) coupled with a photo camera (Nikon, Amstelveen, Netherlands, Digital Sight DS-5M). Samples were evaluated, regarding inflammatory infiltration, fibrosis, angiogenesis, and/or necrosis surrounding the implant, according to ISO-10993-6:2016 guidelines, annex E, by an experienced veterinary pathologist. Scores were attributed to each sample, depending on the individual parameters’ classification, as proposed by the ISO system, enabling a semi-quantitative classification of the implants as “minimal or no reaction” (score 0.0 up to 2.9), “slight reaction” (score 3.0 up to 8.9), “moderate reaction” (score 9.0 up to 15.0) or “severe reaction” (score > 15). As for accessing a systematic biological response, several organs, including lungs, spleen, pancreas, heart, and liver were further histologically analyzed, following a detailed necropsy examination.

### 4.5. Statistical Analysis

A statistical analysis was performed using GraphPad Prism version 6.00 for Mac OS x, GraphPad Software (La Jolla, CA, USA). The experiments were performed in quadruplicates and the results are presented as mean ± standard error of the mean (SE). The analysis was performed by the One-Way ANOVA analysis with the Tukey multi-comparison test. Differences were considered statistically significant at *p* < 0.05. Results significance are presented through the symbol (*), according to the *p*-value, with one, two, three or four symbols, corresponding to 0.01 ≤ *p* < 0.05; 0.001 ≤ *p* < 0.01; 0.0001 ≤ *p* < 0.001; and *p* < 0.0001, respectively.

## 5. Conclusions

Orthopedic impairments related to bone defects represent a major well-being and financial restrain worldwide. Bone tissue engineering has been developing in the past decades, with bioprinting techniques for 3D bone tissue substitutes and metal supporting devices on the treatment’s front line. Overall, this work allowed confirmation of the PLLA coating on SS/SIL devices to be biocompatible (and cytocompatible), as well as the capability to promote osteogenesis, confirming this devices’ potential within bone tissue regenerative techniques. Following these preliminary results, the authors intend to further analyze these devices, considering additional physical surface alterations and in vivo performance in a scale-up animal bone lesion model.

## Figures and Tables

**Figure 1 ijms-22-07655-f001:**
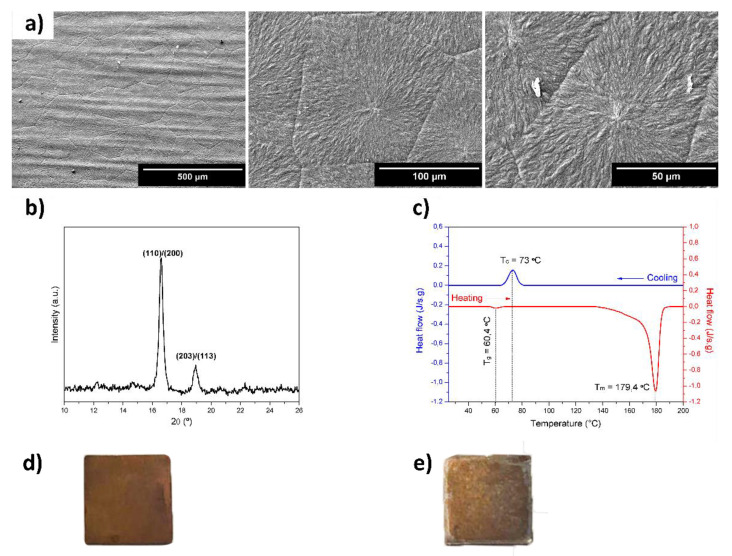
SEM micrographs of PLLA films. The presence of spherulites is noticed and the mean diameter size is 88.9 ± 1.7 μm (**a**); XRD pattern of SS/SIL/PLLA samples with the presence of α-form crystallographic plans (**b**); DSC of PLLA films with the heating and cooling rate of 10 °C/min from room temperature to 200 °C under airflow. T_m_, T_g_, and T_c_ are identified, and the degree of crystallization calculated is 48.8% (**c**). Images of 316SSL devices: (**d**) SS/SIL device and (**e**) SS/SIL/PLLA device.

**Figure 2 ijms-22-07655-f002:**
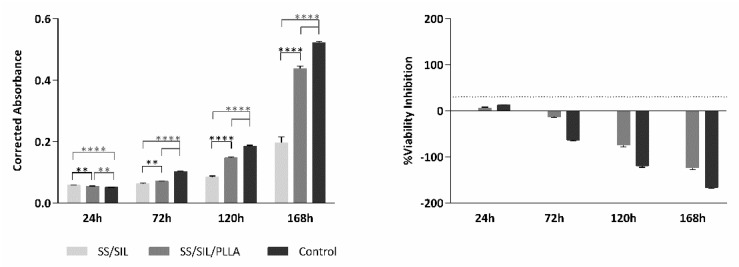
Cytocompatibility assessed by the Presto Blue™ viability assay for hDPSCs. Results are presented in mean ± SE (standard error of the mean). Differences were considered statistically significant at *p* < 0.05. Results significance are presented through the symbol (*), according to the *p*-value, with two or four symbols, corresponding to 0.001 ≤ *p* < 0.01 and *p* < 0.0001, respectively.

**Figure 3 ijms-22-07655-f003:**
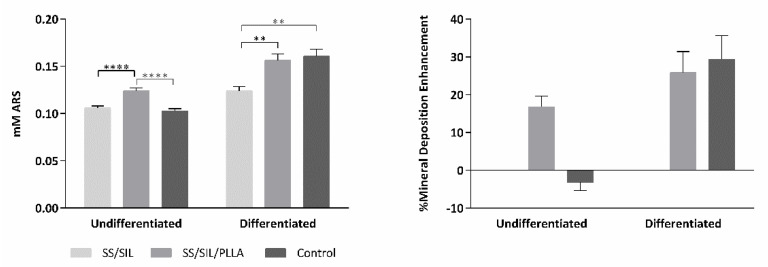
ARS semi-quantification in mM between groups (left panel). Results are presented in mean ± SE. Differences were considered statistically significant at *p* < 0.05. Results significance are presented through the symbol (*), according to the *p*-value, with two or four symbols, corresponding to 0.001 ≤ *p* < 0.01 and *p* < 0.0001, respectively. The % mineral deposition enhancement normalized to the SS/SIL samples are presented in the right panel.

**Figure 4 ijms-22-07655-f004:**
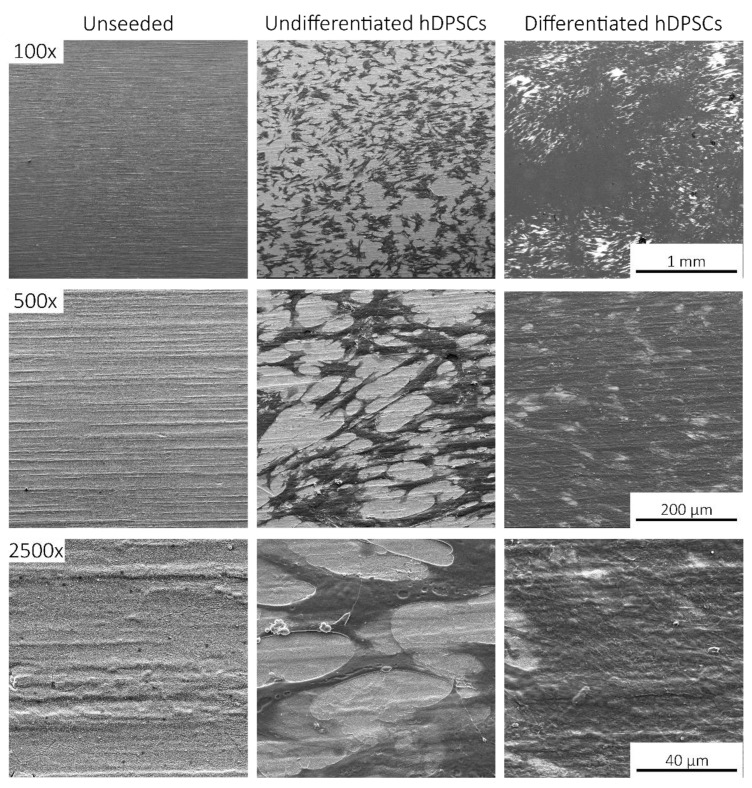
SEM images for the SS/SIL group. Upper panel 100× magnification, middle panel 500× magnification, and lower panel 2500× magnification. First column: Unseeded devices; middle column: Seeded undifferentiated hDPSCs devices; third column: Seeded differentiated hDPSCs devices.

**Figure 5 ijms-22-07655-f005:**
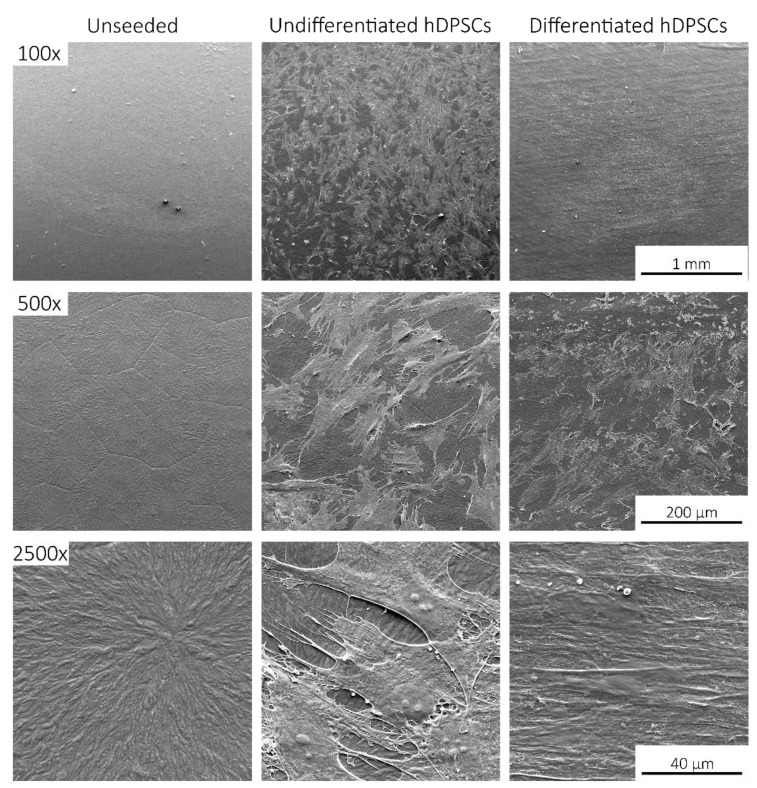
SEM images for the SS/SIL/PLLA group. Upper panel 100× magnification, middle panel 500× magnification, and lower panel 2500× magnification. First column: Unseeded devices; middle column: Seeded undifferentiated hDPSCs devices; third column: Seeded differentiated hDPSCs devices.

**Figure 6 ijms-22-07655-f006:**
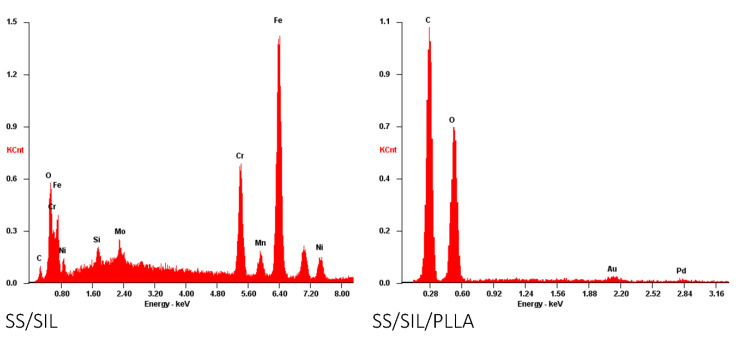
EDS analysis. Left panel: SS/SIL (unseeded sample) and right panel: SS/SIL/PLLA group (unseeded sample).

**Figure 7 ijms-22-07655-f007:**
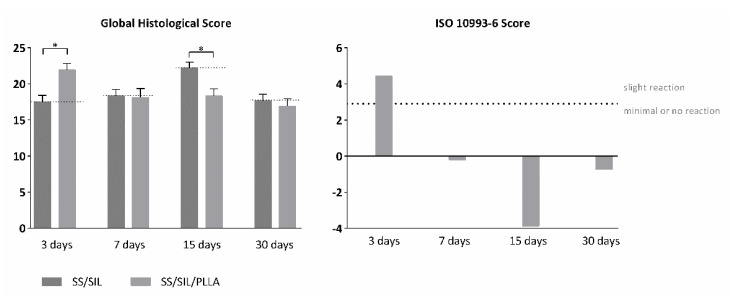
Global histological score (left panel) and calculated score, compared to the negative control (SS/SIL), according to ISO 10993-6:2016 (right panel), for subcutaneous implantation (biocompatibility assessment) of the devices, after 3, 7, 15, and 30 days post-implantation time. Results are presented in mean ± SEM. Differences were considered statistically significant at *p* < 0.05. Results significance are presented through the symbol (*), according to the *p*-value, with one, two, three or four symbols, corresponding to 0.01 ≤ *p* < 0.05; 0.001 ≤ *p* < 0.01; 0.0001 ≤ *p* < 0.001; and *p* < 0.0001, respectively.

**Figure 8 ijms-22-07655-f008:**
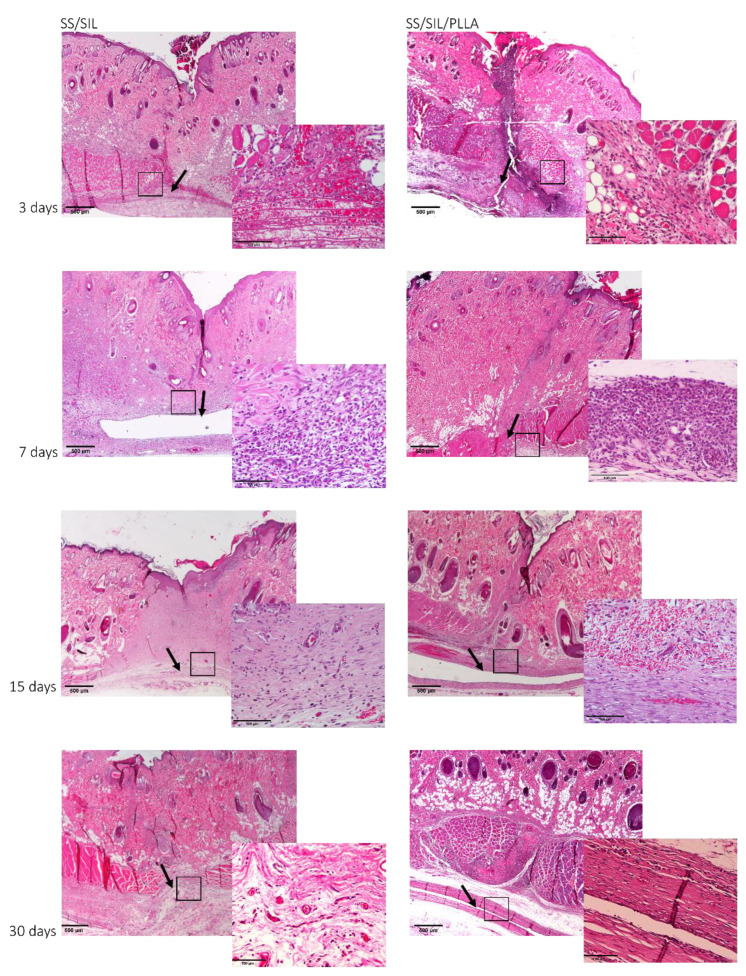
Histological analysis following H&E staining for SS/SIL and SS/SIL/PLLA devices. Evaluation of the sections was performed with a Nikon microscope (Nikon Eclipse E600) coupled with a photo camera (Nikon Digital Sight DS-5M). The left panel corresponds to SS/SIL devices and the right panel to SS/SIL/PLLA devices. For each panel, the left image corresponds to a magnification of 20× (scale bar 500 µm) and the left image to a total magnification of 200× (scale bar 100 µm). Black arrows represent the implantation site and insets indicate 200× magnification images’ acquisition site.

**Figure 9 ijms-22-07655-f009:**
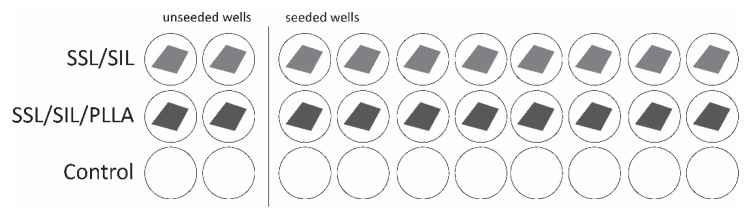
Layout of the experimental set for the in vitro cytocompatibility assessment.

**Table 1 ijms-22-07655-t001:** Cytocompatibility assessed by the Presto Blue™ viability assay for hDPSCs. Corrected absorbance results are presented in mean ± SE.

	SS/SIL		SS/SIL/PLLA		Control	
	Mean	SE	Mean	SE	Mean	SE
24 h	0.058	0.001	0.054	0.001	0.051	0.001
72 h	0.063	0.003	0.071	0.001	0.102	0.001
120 h	0.084	0.004	0.147	0.003	0.185	0.003
168 h	0.196	0.019	0.438	0.008	0.522	0.004

**Table 2 ijms-22-07655-t002:** Cytocompatibility assessed by the Presto Blue™ viability assay for hDPSCs. Results of % viability inhibition are presented in mean ± SE, normalized to the SS/SIL as 100%.

	SS/SIL/PLLA		Control	
	Mean	SE	Mean	SE
24 h	6.317	1.532	12.635	1.024
72 h	−12.948	2.031	−63.214	2.147
120 h	−74.198	3.935	−119.568	3.371
168 h	−123.500	4.059	−166.622	1.934

**Table 3 ijms-22-07655-t003:** ARS semi-quantification in mM between groups. Results are presented in mean ± SE.

	SS/SIL		SS/SIL/PLLA		Control	
	Mean	SE	Mean	SE	Mean	SE
Undifferentiated	0.106	0.002	0.124	0.003	0.103	0.002
Differentiated	0.124	0.004	0.156	0.007	0.161	0.008

**Table 4 ijms-22-07655-t004:** ARS semi-quantification in mM between groups. Results of % mineral deposition enhancement are presented in mean ± SE, normalized to the SS/SIL as 100%.

	SS/SIL/PLLA		Control	
	Mean	SE	Mean	SE
Undifferentiated	16.789	2.917	−3.141	2.258
Differentiated	25.851	5.553	29.384	6.203

**Table 5 ijms-22-07655-t005:** Global histological scores presented as mean + SE for both devices, and ISO-10993-6 score at 3, 7, 15, and 30 days after implantation.

	SS/SIL		SS/SIL/PLLA	
	Mean	SE	Mean	SE
3 days	17.50	0.909	21.93	0.892
	ISO SCORE		4.43	
7 days	18.33	0.882	18.13	1.230
	ISO SCORE		−0.20	
15 days	22.20	0.827	18.33	0964
	ISO SCORE		−3.87	
30 days	17.67	0.908	16.93	1.021
	ISO SCORE		−0.74	

## Data Availability

The data that support the findings of this study are available from the corresponding author on request.

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
