# Peer review of "In Vitro and In Vivo Characterization of PLLA-316L Stainless Steel Electromechanical Devices for Bone Tissue Engineering—A Preliminary Study"

_ijms, 2021, doi:10.3390/ijms22147655_

Round 1

Reviewer 1 Report

This manuscript examined the potential of PLLA-SS316L device for bone tissue regeneration in vitro and in vivo. The authors suggested that PLLA-SS316L device had osteoconductive properties in vitro and had biocompatibility in vivo.

The manuscript is well written and the results are conclusive data. However, I have the following suggestions to the authors with further revision for the manuscript.

  1. The authors should show the SS/SIL/PLLA device.
  2. The author describes why hDPSCs was used in this study line 310-321. However, the authors should use bone lineage cells derived from mesenchymal stem cells like as bone marrow and so on. Thus, the authors should clarify the rationale for using DPSC.
  3. In Fig.8, The authors should clarify where the devices were inserted.

Author Response

Reviewer N. 1

This manuscript examined the potential of PLLA-SS316L device for bone tissue regeneration in vitro and in vivo. The authors suggested that PLLA-SS316L device had osteoconductive properties in vitro and had biocompatibility in vivo.

The manuscript is well written and the results are conclusive data. However, I have the following suggestions to the authors with further revision for the manuscript.

  1. The authors should show the SS/SIL/PLLA device.

As suggested by the Reviewer, authors have made changes in Figure 1, and included images of both devices.

  1. The author describes why hDPSCs was used in this study line 310-321. However, the authors should use bone lineage cells derived from mesenchymal stem cells like as bone marrow and so on. Thus, the authors should clarify the rationale for using DPSC.

The authors acknowledge the Reviewers’ question and made further changes in the discussion section. The authors believe hDPSCs to be a proven source of MSCs for bone tissue regeneration studies, even when compared to bone-marrow derived MSCs. Several studies have been comparing both cell populations for bone regeneration, and overall, DPSCs potential has been established and accepted (https://doi.org/10.1002/term.220, https://doi.org/10.3727/096368910X539128) as a promising cell source. By using hDPSCs in this study, the authors intended not only to use a cell population with promising potential towards mineralized tissue regeneration in in vitro studies, but also to further investigate these cells’ ability in in vivo studies and in further clinical application, themselves and/or in combination with their secretory products, as an alternative and non-invasive and easily accessible (due to cryopreservation banks) source of mesenchymal stem/stromal cells, prospecting to further clinical application scenarios.

The following paragraph was included in the discussion section:

“The validity of these cells’ application for in vitro and in vivo studies towards bone tissue regeneration has been previously established and accepted [43, 44]. Furthermore, envisioning clinical application, they represent a non-invasive and easily accessible cell source [26].”

  1. In Fig.8, The authors should clarify where the devices were inserted.

As suggested by the Reviewer, authors have made changes in Figure 8, and included arrows for each image, showing where the devices were implanted.

Changes to the Manuscript

  • Figures improved: 1 and 8 (and legend)
  • Results (2.1)
  • Discussion section
  • References

Reviewer 2 Report

" In vitro and in vivo characterization of PLLA-316L stainless steel electromechanical devices for bone tissue engineering – a preliminary study"

It is very interesting to focus to investigate in vitro and in vivo characterization of PLLA-316L stainless steel electromechanical devices for bone tissue engineering. The manuscript is well written. However, there are a few corrections that are essential to meet the standard for publication. Please refer to the following comments.

1) Your current study shows that it is a bone tissue regeneration technique that promotes bone formation. You had proposed a research topic in an animal bone lesion model as the next study.

Please add your opinion in the discussion section on the clinical future prospects of your study.

2) In Figure 8, please clarify which part of the 20x HE-stained tissue image the 200x magnified image shows. Please indicate the enlarged part with a square.

3) Please correct the sentence as a whole to a paragraph-separated sentence. The content is so great that a reader-friendly paper will increase the citations of your research paper.

Author Response

Reviewer N. 2

" In vitro and in vivo characterization of PLLA-316L stainless steel electromechanical devices for bone tissue engineering – a preliminary study"

It is very interesting to focus to investigate in vitro and in vivo characterization of PLLA-316L stainless steel electromechanical devices for bone tissue engineering. The manuscript is well written. However, there are a few corrections that are essential to meet the standard for publication. Please refer to the following comments.

1) Your current study shows that it is a bone tissue regeneration technique that promotes bone formation. You had proposed a research topic in an animal bone lesion model as the next study.

Please add your opinion in the discussion section on the clinical future prospects of your study.

According to the Reviewers comment, the following paragraph was added to the discussion section:

“Non-critical and critical bone defect will be considered, on the femur and on the iliac crest, respectively, according to previous bone lesion models [3, 25]. Devices will be prepared as screws and plates. The authors believe these devices, along with further improvements to their surface, to present strong regenerative characteristics towards bone lesion stabilization and support.”

2) In Figure 8, please clarify which part of the 20x HE-stained tissue image the 200x magnified image shows. Please indicate the enlarged part with a square.

As suggested by the Reviewer, authors have made changes in Figure 8.

3) Please correct the sentence as a whole to a paragraph-separated sentence. The content is so great that a reader-friendly paper will increase the citations of your research paper.

The authors would like to ask the Reviewer to clarify which sentence the suggestion refers to, as to proceed with the changes proposed.

Changes to the Manuscript

  • Figures improved: 1 and 8 (and legend)
  • Results (2.1)
  • Discussion section
  • References

Reviewer 3 Report

Thanks for submitting this manuscript.
I have carefully read your manuscript with great interest.

However, I confused to understand some points.

Major comments

  1. 2 paragraph need to modify. This paragraph is hard to understanding.
  2. Figure 2 Compare to Corrected absorbance values, % of viability inhibition looks like huge different in 72hr. Why is it?
  3. the SS/SIL/PLLA group can be classified as non-cytotoxic, as the % of viability inhibition never exceeded the 30% pre-established limit throughout the assay, according to the upper mentioned guideline -> Please don’t use “never”. Have you tested it every time? After overnight, this assay can be used.
  4. Every p value: 0.01 < p ≤ 0.05;001 < p 0.01; 0.0001 < p ≤ 0.001 and p ≤ 0.0001 is wrong.
  5. As shown in the %Viability Inhibition graph in figure 3 (lower panel)-> Maybe Figure 2 right panel??, the SS/SIL/PLLA group can be classified as non-cytotoxic, as the % of viability inhibition never exceeded the 30% pre-established limit throughout the assay, according to the upper mentioned guideline.
  6. 3 paragraph. Please explain why ARS solution protocol used
  7. Line199 hDPSCs were employed?? for this assay
  8. Ling199 Results demonstrate the PLLA coated devices to enhance mineral deposition in the group supplemented with osteodifferentiation media, when comparing to the SS/SIL devices, hence suggesting PLLA coating to promote osteodifferetiation -> Maybe Figure 3 left? Please check this sentence.
  9. Please explain and describe why you did the Figure 3 right experiment and why only compare SS/SIL/PLLA and control. I thought you would compare between SS/SIL and SS/SIL/PLLA.
  10. How to define undifferentiated or differentiated cells? (Figure 3.)
  11. Line 211. The authors have chosen not to normalize the results to the same area, as they believe it would insert bias into the data. -> Does that mean it will be possible to select data?? How can I trust other data? Figure 3. Right and left is very difference.
  12. figures 4, 5 and 6 -> Figures
  13. 5 paragraph. -> So what? What do you want to say? Author need to detail explain.
  14. Line 322. According to the results obtained, the SS/SIL/PLLA devices presented superior viability outcomes, thus suggesting the PLLA coating to enhance cell proliferation and viability -> compare to what?
  15. Line 325. Due to the non-transparent nature of the devices, a morphologic qualitative assessment was performed by SEM, with both devices (SS/SIL and SS/SIL/PLLA) presenting cellular monolayer formation, with normal cell morphology and attachment (Figures 4 and 5). -> need the control experiment.

Author Response

Reviewer N. 3

Thanks for submitting this manuscript.
I have carefully read your manuscript with great interest.

However, I confused to understand some points.

Major comments

  1. 2 paragraph need to modify. This paragraph is hard to understanding.

The authors understand the reviewers’ comment. The following changes were applied to the paragraph: “The Presto BlueTM viability assay was performed in SS/SIL (gold standard) and SS/SIL/PLLA samples. A group with seeded cells but with no biomaterial was considered as a control, of the cell population health and normal behaviour in culture, growth, and proliferation. Corrected absorbance values were obtained for each time-point (24, 72, 120 and 168 h) and are presented in Figure 2 (left panel) and Table 1. The % of viability inhibition, normalized to the SS/SIL group, is presented on the right panel of Figure 2 and Table 2.”

  1. Figure 2 Compare to Corrected absorbance values, % of viability inhibition looks like huge different in 72hr. Why is it?

The authors understand the reviewers’ comment. Data and graphs were again checked and are correct and according to data obtained. When normalizing values to a control and presenting them in %, some differences become more evident, and that could be the case of the values at 72h.  From the left panel to the right panel, data were normalized to the SS/SIL devices’ mean values and presented in %.

  1. the SS/SIL/PLLA group can be classified as non-cytotoxic, as the % of viability inhibition never exceeded the 30% pre-established limit throughout the assay, according to the upper mentioned guideline -> Please don’t use “never”. Have you tested it every time? After overnight, this assay can be used.

The authors understand the reviewer’s comment. This assay was taken at the established timepoints, and with 1 hour incubation, according to manufacturing instructions. At every timepoint, the samples did not exceed the 30% viability inhibition limit established by the ISO 10993-5 guideline. The authors have made the following changes to the sentence, as suggested by the reviewer: “As shown in the %Viability Inhibition graph in figure 3 (right panel), the SS/SIL/PLLA group can be classified as non-cytotoxic, as the % of viability inhibition did not exceed the 30% pre-established limit throughout the assay, according to the upper mentioned guideline”.

  1. Every p value: 0.01 < p ≤ 0.05;001 < p 0.01; 0.0001 < p ≤ 0.001 and p ≤ 0.0001 is wrong.

The authors have corrected the P values along the document.

  1. As shown in the %Viability Inhibition graph in figure 3 (lower panel)-> Maybe Figure 2 right panel??, the SS/SIL/PLLA group can be classified as non-cytotoxic, as the % of viability inhibition never exceeded the 30% pre-established limit throughout the assay, according to the upper mentioned guideline.

The authors have made the following changes to the sentence: “As shown in the %Viability Inhibition graph in Figure 3 (right panel), the SS/SIL/PLLA group can be classified as non-cytotoxic, as the % of viability inhibition did not exceed the 30% pre-established limit throughout the assay, according to the upper mentioned guideline”.

  1. 3 paragraph. Please explain why ARS solution protocol used

The authors have selected the ARS solution protocol for the osteogenic differentiation assessment, due to this method versatility, simplicity and reliability. ARS has been extensively used for this purpose along the scientific community (doi:10.1016/j.ab.2004.02.002) and in previous works from the authors group (https://doi.org/10.1155/2020/2938258 and https://doi.org/10.1371/journal.pone.0203936).

  1. Line199 hDPSCs were employed?? for this assay

The authors have made the following changes to the sentence: “hDPSCs were used for this assay, as for the cytocompatibility assessment.”

  1. Ling199 Results demonstrate the PLLA coated devices to enhance mineral deposition in the group supplemented with osteodifferentiation media, when comparing to the SS/SIL devices, hence suggesting PLLA coating to promote osteodifferetiation -> Maybe Figure 3 left? Please check this sentence.

The authors have made the following changes to the sentence: “Results shown in Figure 3 demonstrate the PLLA coated devices to enhance mineral deposition in the group supplemented with osteodifferentiation media, when comparing to the SS/SIL devices, hence suggesting PLLA coating to promote osteodifferentiation.”

  1. Please explain and describe why you did the Figure 3 right experiment and why only compare SS/SIL/PLLA and control. I thought you would compare between SS/SIL and SS/SIL/PLLA.

The authors understand the reviewer’s comment. With the right panel in Figure 3 the authors intended to present the mineral deposition enhancement obtained on the SS/SIL/PLLA and control samples, when normalizing to the SS/SIL samples, that is considered the gold standard.  The statistical analysis was performed comparing to the SS/SIL devices, as shown on the left panel in Figure 3, thus, panel in the right is only to evidence the SS/SIL/PLLA samples mineral deposition enhancement, when normalized to the gold standard group.

The authors have made the following changes to the Figure 3 legend: “Figure 3. ARS Semi-Quantification in mM between groups (left panel). Results presented in Mean ± SE. Differences were considered statistically significant at P < 0.05. Results significance are presented through the symbol (*), according to the p value, with one, two, three or four symbols, corresponding to 0.01 ≤ P < 0.05; 0.001 ≤ P < 0.01; 0.0001 ≤ P < 0.001 and P < 0.0001, respectively. % Mineral deposition enhancement normalized to the SS/SIL samples presented in right panel.”

  1. How to define undifferentiated or differentiated cells? (Figure 3.)

With this assay (ARS), differentiation is analysed by measuring the ECM content in mineral deposition (calcium deposits). The ARS protocol uses a dye that is easily extracted from the stained cell monolayer and measures the mineral content of the ECM by colorimetric detection at 405 nm. In this method, differentiation is assessed by the extent of mineral deposition in the ECM. This can occur in a pro-osteogenic differentiation condition (differentiated group) and it can also occur spontaneously on a normal culture condition (undifferentiated group).

  1. Line 211. The authors have chosen not to normalize the results to the same area, as they believe it would insert bias into the data. -> Does that mean it will be possible to select data?? How can I trust other data? Figure 3. Right and left is very difference.

The authors understand the reviewer’s comment. The authors could have normalized the values of the absorbances to the areas of the surface where the cells were attached. Devices presented slightly smaller areas then the well bottom (control group). If this normalization had been made, differences between SS/SIL and SS/SIL/PLLA would be the same, as they have the same area. However, differences between the control group and the SS/SIL and SS/SIL/PLLA samples would have been attenuated. The authors have chosen not to proceed with this normalization, as they believe bias would have been introduced in the data by doing so. For a correct normalization, cell confluency would have to be taken into consideration, and the non-transparent nature of the devices did not allow for such an evaluation. Furthermore, this control group was considered a control of the cellular population normal behaviour and health in culture, and not a test group to be compared to the SS/SIL/PLLA samples directly. The authors believe this decision would not affect to any extent the conclusions obtained in this work regarding SS/SIL and SS/SIL/PLLA devices.

  1. figures 4, 5 and 6 -> Figures

The authors have made the following changes to the sentence: “The results are presented in Figures 4, 5 and 6.”

  1. 5 paragraph. -> So what? What do you want to say? Author need to detail explain.

The authors have made the following changes to the paragraph: “Macroscopical evaluation of the subcutaneous tissue revealed no infection, inflammation, nor hemorrhage. The scoring system was obtained by subtracting the global histological score of the SS/SIL devices for each timepoint. Overall, microscopically samples presented minimal fibrosis and a broad brand of capillaries with supporting structures, at every timepoint (Figure 8). Polymorphonucleated (PMN) cells presence was detected at each timepoint for both samples but decreased along with the recovery period. Necrosis, as well as giant cells were only seldomly identified. Mononuclear inflammatory cells were greatly detected, when compared to PMN cells, for all groups and recovery periods, except for the 3 days timepoint. According to ISO 10993-6 scoring system, the SS/SIL/PLLA device was classified as “slight reaction” at 3 days post-implantation time, and as “minimal to no reaction” at 7, 15 and 30 days post-implantation time, comparing to the gold standard SS/SIL devices. No relevant alterations were noted in the microscopical evaluation of the different organs. Considering the previous mentioned guideline, SS/SIL/PLLA devices can be classified as biocompatible.”

  1. Line 322. According to the results obtained, the SS/SIL/PLLA devices presented superior viability outcomes, thus suggesting the PLLA coating to enhance cell proliferation and viability -> compare to what?

The authors have made the following changes to the sentence: “According to the results obtained, the SS/SIL/PLLA devices presented superior viability outcomes, thus suggesting the PLLA coating to enhance cell proliferation and viability, when comparing with the uncoated SS/SIL devices.”

  1. Line 325. Due to the non-transparent nature of the devices, a morphologic qualitative assessment was performed by SEM, with both devices (SS/SIL and SS/SIL/PLLA) presenting cellular monolayer formation, with normal cell morphology and attachment (Figures 4 and 5). -> need the control experiment.

The authors understand the reviewer’s comment. SEM images were acquired for the control group, seeded on the well bottom. The authors have chosen not to present these data for this paper, as they believe imagens will not provide additional relevant data for this work. The following layout has been prepared for the Reviewer:

Figure. SEM images for the control group. Upper panel 100x magnification, middle panel 1000x magnification and lower panel 2000x magnification. First column: unseeded devices; middle column: seeded undifferentiated hDPSCs devices; third column: seeded differentiated hDPSCs devices.

Round 2

Reviewer 3 Report

I recognize that some work has been done by the authors to satisfy my requests and compensate for the shortcomings and confusion present in the original version, however even in this form the manuscript is still a long way from being publishable.

Line 177: “As shown in the %Viability Inhibition graph in Figure 3 (right panel), the 177 SS/SIL/PLLA group can be classified as non-cytotoxic, as the % of viability inhibition did 178 not exceed the 30% pre-established limit throughout the assay”

Why the authors did not revise “As shown in the %Viability Inhibition graph in Figure 3 (right panel)”?

Figure 3 (right panel) present %Mineral Deposition enhancement.

Line 213: “The authors have chosen not to normalize the results to the same area, as they believe it would insert bias into the data. Furthermore, normalization also depends on cell confluency, and the non-transparent nature of the samples impaired such observations and further qualitative assessment.”

I don’t agree this decision. I believe that author’s decision are occurred to insert bias into data.

How the authors estimated “normalization also depends on cell confluency”? what scientific evidence?

Author Response

Reviewer 3 – round 2

“I recognize that some work has been done by the authors to satisfy my requests and compensate for the shortcomings and confusion present in the original version, however even in this form the manuscript is still a long way from being publishable.”

  • Line 177: “As shown in the %Viability Inhibition graph in Figure 3 (right panel), the 177 SS/SIL/PLLA group can be classified as non-cytotoxic, as the % of viability inhibition did 178 not exceed the 30% pre-established limit throughout the assay”

Why the authors did not revise “As shown in the %Viability Inhibition graph in Figure 3 (right panel)”?

Figure 3 (right panel) present %Mineral Deposition enhancement.

Answer:

Corrections were made in line 177:

“As shown in the %Viability Inhibition graph in Figure 2 (right panel)”

Figure 2 was by mistake referred to as Figure 3 in this line.

  • Line 213: “The authors have chosen not to normalize the results to the same area, as they believe it would insert bias into the data. Furthermore, normalization also depends on cell confluency, and the non-transparent nature of the samples impaired such observations and further qualitative assessment.”

I don’t agree this decision. I believe that author’s decision are occurred to insert bias into data.

How the authors estimated “normalization also depends on cell confluency”? what scientific evidence?

Answer:

The authors understand the Reviewers comment. Considering the experimental setup, cells were seeded on 1cm2 samples, for the devices’ groups, and 1.9 cm2 for the control group (24 well plate bottom standard dimensions). Thus, attachment area for cell growth is not equal between devices’ groups and the control group. However, between both devices the area is the same and can be directly compared. The authors, along this work, did not take any conclusions on the comparison between the devices and the control group. They refer to this control group as a control of the cell population normal behaviour in culture, proliferation, and health. Comparison and conclusions are taken between both devices’ groups, being the SS/SIL and the SS/SIL/PLLA, and both having the same area, and thus needing no normalization. The authors strongly believe bias would be inserted to the data, by changing the area of the control group. Data is presented with no normalization to area; thus no bias is inserted.  Paragraph has been revised and changed, as to avoid further confusion.
